# Imaging and Hemodynamic Characteristics of Vulnerable Carotid Plaques and Artificial Intelligence Applications in Plaque Classification and Segmentation

**DOI:** 10.3390/brainsci13010143

**Published:** 2023-01-13

**Authors:** Na Han, Yurong Ma, Yan Li, Yu Zheng, Chuang Wu, Tiejun Gan, Min Li, Laiyang Ma, Jing Zhang

**Affiliations:** 1Department of Magnetic Resonance, Lanzhou University Second Hospital, Lanzhou 730030, China; 2Gansu Province Clinical Research Center for Functional and Molecular Imaging, Lanzhou 730030, China; 3Second Clinical School, Lanzhou University, Lanzhou 730030, China; 4School of Mathematics and Statistics, Lanzhou University, Lanzhou 730030, China

**Keywords:** vulnerable plaque, VW-HRMRI, 4D flow, artificial intelligence, stroke

## Abstract

Stroke is a massive public health problem. The rupture of vulnerable carotid atherosclerotic plaques is the most common cause of acute ischemic stroke (AIS) across the world. Currently, vessel wall high-resolution magnetic resonance imaging (VW-HRMRI) is the most appropriate and cost-effective imaging technique to characterize carotid plaque vulnerability and plays an important role in promoting early diagnosis and guiding aggressive clinical therapy to reduce the risk of plaque rupture and AIS. In recent years, great progress has been made in imaging research on vulnerable carotid plaques. This review summarizes developments in the imaging and hemodynamic characteristics of vulnerable carotid plaques on the basis of VW-HRMRI and four-dimensional (4D) flow MRI, and it discusses the relationship between these characteristics and ischemic stroke. In addition, the applications of artificial intelligence in plaque classification and segmentation are reviewed.

## 1. Introduction

AIS is the second leading cause of death to humans after cancer. High recurrence, disability and mortality pose heavy economic and disease burdens on patients and their families [1]. Embolism caused by carotid atherosclerotic plaque shedding has been recognized as the major cause of ischemic stroke worldwide, accounting for approximately 18% to 25% of ischemic strokes [2,3,4]. A prospective longitudinal followed-up study showed that the risk of stroke increased with carotid plaque vulnerability characteristics [5].

Vulnerable carotid plaques have specific morphological and hemodynamic characteristics. However, conventional angiography techniques, such as digital subtraction angiography (DSA), computed tomography angiography (CTA) and magnetic resonance angiography (MRA), can assess only the degree of vascular stenosis. They cannot provide specific characteristics of vulnerable carotid plaques. However, guidelines from the American Society of Neuroradiology (ASNR) have highlighted that not only the degree of vascular stenosis but also plaque composition is attributable to the risk of stroke associated with carotid plaques [6]. Developments in imaging techniques have enabled the detecting of the characteristics of carotid plaque vulnerability. VW-HRMRI can not only assess vascular stenosis but also characterize plaque morphology and composition and identify vulnerable plaques. Moreover, studies have revealed that whether vulnerable plaques progress or even rupture is not only dependent on plaque morphology and composition but also subject to hemodynamic factors exerted on the surface of plaques. Four-dimensional flow enables the measurement of hemodynamic parameters along any direction, from any angle and at any time in the cardiac cycle, which can comprehensively and carefully reflect changes in carotid plaque hemodynamics [7]. Therefore, VW-HRMRI combined with 4D flow shows great clinical value for accurately identifying vulnerable carotid plaques. Moreover, in recent years, artificial intelligence has been increasingly applied in clinical practice and contributed to advances in this area for atherosclerotic plaques research.

This review summarizes developments in the imaging and hemodynamic characteristics of vulnerable carotid plaques based on VW-HRMRI and 4D flow, and it discusses the relationship between these characteristics and ischemic stroke. In addition, the applications of artificial intelligence in plaque classification and segmentation are reviewed.

## 2. Imaging Characteristics of Vulnerable Carotid Plaques

### 2.1. Qualitative Analysis of Vulnerable Carotid Plaques Based on VW-HRMRI

Vulnerable carotid plaques have the following histopathologic features: lipid-rich necrotic cores (LRNC, >40% of plaque volume), thin fibrous cap (<165 μm), intraplaque hemorrhage (IPH), intraplaque inflammation and neovascularization, plaque surface ulceration and positive vascular remodeling. The ability of VW-HRMRI to classify plaques has been demonstrated by evidence of high consistency between VW-HRMRI and histopathology in detecting vulnerable plaques [8]. Plaque components have characteristic imaging findings on VW-HRMRI and the imaging characteristics of carotid vulnerable plaques and stable plaques, as shown in Figure 1 and Figure 2.

#### 2.1.1. Lipid-Rich Necrotic Core and Thin Fibrous Cap

Plaque vulnerability is associated with the LRNC and thin fibrous cap. LRNC is composed mainly of cholesterol crystals, apoptotic cell debris and calcium, showing isointensity on time-of-flight (TOF) images, T1-weighted images (T1WIs) and T2-weighted images (T2WIs), with no or mild enhancement [6,9,10]. The lipid core progress in atherosclerosis that was a result of proliferative monocytes and smooth muscle cells increased lipid ingestion and then increased phagocytes or foam cells that gradually disintegrate to cause intercellular lipid accumulation [11].

LRNC has been shown to be an independent imaging predictor of vulnerable carotid plaques, with the area under the receiver operating characteristic curve (AUC) used for identifying vulnerable plaques being 0.690 [12]. The lipid core ratio quantifies the amount of lipid in a plaque. Its calculation formula is as follows: lipid core ratio = lipid core area/vascular wall area. Semiautomatic plaque analysis software (MRI-Plaque View, VesselMass) that is based on VW-HRMRI can account for quantifying plaque volume and the percentage of LRNC, IPH, calcification, etc. The LRNC can increase the risk of stroke. A longitudinal study of 120 asymptomatic individuals revealed that carotid plaques with a maximum percentage of LRNC greater than 40% are at high risk and prone to fibrous cap rupture during a 3-year follow-up [13].

The fibrous cap is a layer of fibrous connective tissue covering the lipid core. An intact fibrous cap presents as a continuous hypointensity band on TOF, often isointensity on T1WIs, hypointensity or hyperintensity on T2WIs and slight hyperintensity on proton density-weighted images (PDWIs). Fibrous cap thinning or discontinuity is an important characteristic of vulnerable plaque. Gadolinium-based contrast enhancement VW-HRMRI is the imaging technique of choice for imaging this characteristic [14].

#### 2.1.2. Intraplaque Hemorrhage

IPH is one of the key characteristics of vulnerable plaques and is mainly attributable to vascular endothelial damage and neovascular dilation and leakage [15]. The AUC of IPH is used for identifying vulnerable plaques as 0.708 higher than LRNC, but the combination of IPH and LRNC improved the AUC to 0.825 [12]. In addition, among characteristics that can be detected in vulnerable plaques, a meta-analysis of nine studies has demonstrated that IPH has a strong association with the increased occurrence of future ischemic stroke in patients with symptomatic and asymptomatic carotid stenosis. Compared with asymptomatic carotid plaques, symptomatic carotid plaques tend to contain a larger volume of IPH, and the volume of IPH is independently associated with AIS in patients with symptomatic carotid plaques, suggesting that the expansion of IPH volume is associated with an increased occurrence of AIS [16,17]. Therefore, carotid revascularization is required in patients with symptomatic carotid stenosis whose IPH has been found. In addition, a longitudinal followed-up MRI investigation of 1190 people with asymptomatic carotid stenosis suggests that those with IPH have considerably lower event-free survival than people without IPH [5]. The underlying mechanism may be that IPH precipitates the risk of atherosclerotic plaque progression and rupture. In the pathophysiological process of IPH, phagocytosis releases hemoglobin from red blood cells, which promotes local inflammation and activates proteolytic enzymes, thereby accelerating the degradation and destruction of fibrous caps [18]. Another long-term longitudinal MRI follow-up study of 198 carotid plaque patients reported an increase in the prevalence of IPH with age and hypertension, and it underlined the significance of decreasing blood pressure to avoid stroke [17].

The signal of IPH varies because it depends on the oxidative state of hemoglobin. VW-HRMRI is the best imaging technique for the detection of IPH. It shows up as hyperintensity on T1WIs and isointensity or hypointensity on T2WIs around week 1; hyperintensity on both T1WIs and T2WIs during weeks 1–6; and hypointensity on both T1WIs and T2WIs after week 6. However, no evidence is available to connect the time of IPH with occurrence of AIS; thus, further studies are needed.

#### 2.1.3. Intraplaque Inflammation and Neovascularization

Other important characteristics of vulnerable plaques are intraplaque inflammation and neovascularization. Studies have shown that ¹⁸F-sodium fluoride PET and molecular imaging are innovative techniques for the detection of plaque inflammation, and contrast-enhanced ultrasound (US) and computed tomography (CT) can also be applied to detect intraplaque neovascularization [19,20]. However, MRI is superior to the above techniques because this technique has no ionizing radiation but high soft-tissue contrast and is convenient for clinical application. Therefore, contrast-enhanced VW-HRMRI is the preferred imaging technique for the detection of intraplaque inflammation and neovascularization. With the pituitary stalk as the reference [21], hyper- or iso-enhancement relative to the pituitary stalk indicates inflammatory cell infiltration and neovascularization in plaques, and inflammatory cells typically accumulate in the fibrous cap of the plaque [22].

A cross-sectional study of 62 patients with carotid atherosclerotic plaques reported that intraplaque macrophage infiltration is associated with plaque rupture and AIS [23], but the relationship between intraplaque neovascularization and stroke is inconclusive at present.

#### 2.1.4. Plaque Surface Ulceration

The surface of carotid plaques can be classified as smooth or irregular. Irregularity in the plaque surface—particularly ulceration—is considered as a characteristic of vulnerable plaques. The definition of carotid plaque ulceration varies depending on the modality used or even among different research groups. In terms of histology, ulceration describes an endothelial defect of at least 1000 μm in width, resulting in the exposure of the plaque’s necrotic core to circulation. On an image, ulceration is reserved for cavities measuring at least 1 mm or 2 mm, according to different studies and proposed risk stratification systems [24,25]. The carotid plaque surface can be assessed by using contrast-enhanced US, CT and MRI, with varying levels of diagnostic accuracy. Two cross-sectional studies have shown that diagnostic accuracy for detecting ulcers by using contrast-enhanced CT and contrast-enhanced MRI were superior to that with contrast-enhanced US [26,27].

One study demonstrated that ulceration was associated not only with the presence of IPH but also with LRNC. It is a precursor of stroke events, and ulcerated plaques are related to a sevenfold increase in ipsilateral stroke risk [28]. Animal studies and clinical trials such as NASCET have shown that the dents in the plaque surface ulceration create a vortex flow that promotes platelet aggregation, resulting in an increased risk of future stroke through a mechanism such as artery-to-artery embolism [29].

#### 2.1.5. Positive Vascular Remodeling

Vulnerable carotid plaques can also be identified on the basis of the pattern of vascular remodeling. During the slow formation of atherosclerotic plaques, the vascular lumen can develop positive remodeling (wall thickening toward the outside of the lumen) and negative remodeling (wall thickening toward the inside of the lumen). Plaques with negative remodeling typically have small lipid cores with calcification and fibrosis, indicating a stable state of plaques. Those with positive remodeling are usually characterized by LRNC and intraplaque inflammation, which are vulnerable plaques prone to rupture [30]. One study showed that vascular stenosis is not an independent predictor of AIS, and as many as 27% of patients with fatal AIS had only mild to moderate vascular stenosis [31], which could be explained by the positive vascular remodeling of vulnerable carotid plaques. In addition, Qiao et al. [32] found that the arteries in the posterior circulation are more likely to develop positive vascular remodeling.

### 2.2. Quantitative Analysis of Vulnerable Carotid Plaques Based on VW-HRMRI

In addition to the above qualitative imaging characteristics, the identification of vulnerable carotid plaques also includes quantitative imaging characteristics. Common quantitative parameters include plaque volume and thickness, the normalized wall index (NWI), the remodeling index and the vascular stenosis ratio, as shown in Table 1.

#### 2.2.1. Plaque Volume and Thickness

Larger plaque volume is a useful characteristic for identifying vulnerable carotid plaques. Similar to plaque volume, the large maximum plaque thickness is a characteristic of vulnerable carotid plaques because it is linked to the large plaque volume. MRI possesses superior soft-tissue contrast, so VW-HRMRI is highly useful for quantifying the thickness and volume of carotid plaques.

A cross-sectional study of 1072 patients showed that larger maximum carotid plaque thickness (quantified by MRI) was strongly associated with cerebral ischemic symptoms and the occurrence of stroke [33]. One retrospective study evaluated the change of carotid artery plaque volume and found larger plaque volume to be associated with vulnerable plaques and stroke [34]. In addition, one prospective longitudinal study indicated that the progressive increase in carotid plaque volume (quantified by MRI) was associated with ischemic stroke [35].

#### 2.2.2. Plaque Normalized Wall Index

NWI based on VW-HRMRI is a better indicator to evaluate carotid plaque burden than the degree of vascular stenosis, with high accuracy and repeatability of measurements. In a study of middle cerebral artery atherosclerosis, measures of plaque burden significantly improved the assessment of vulnerable plaques compared with the degree of vascular stenosis [36]. NWI = wall area (WA)/[lumen area (LA) + wall area (WA)]. The normal carotid artery NWI is approximately 0.4, and a higher NWI (>0.56) has been shown to be associated with LRNC, IPH and fibrous cap rupture [37]. NWI has also been demonstrated as a useful indicator to assess the severity of AIS in patients with carotid atherosclerosis, showing that patients with higher National Institutes of Health Stroke Scale (NIHSS) scores have higher NWI values [38].

#### 2.2.3. Plaque Remodeling Index and Vascular Stenosis Ratio

The plaque remodeling index is a quantitative indicator to evaluate the pattern of vascular remodeling. Its calculation formula is as follows: remodeling index = vascular area at the slice with the narrowest lumen/vascular area at the adjacent slice with normal vascular wall. A remodeling index ≥1.0 indicates positive remodeling, while a remodeling index <1.0 indicates negative remodeling.

The degree of vascular stenosis is assessed by the stenosis ratio defined by NASCET (North American Symptomatic Carotid Endarterectomy Trial) criteria. Stenosis ratio = (1 − [lumen area at the slice with the narrowest lumen/lumen area at the adjacent slice with normal vascular wall]) × 100%. Numerous studies have shown that the degree of vascular stenosis is an influencing factor for AIS but is insufficient to predict the occurrence of AIS in patients with carotid atherosclerosis, because in most cases, vulnerable carotid plaques are accompanied by positive vascular remodeling [39]. A growing body of research indicates that even in cases of mild or moderate vascular stenosis, vulnerable carotid plaques can still cause ischemic stroke [40,41].

## 3. Hemodynamic Characteristics of Vulnerable Carotid Plaques

Morphology and hemodynamics are the focuses of the study of carotid atherosclerotic plaques. As mentioned above, carotid plaque morphology studies have been considerably advanced with the application of VW-HRMRI. However, by morphology alone, it still cannot completely distinguish vulnerable plaques from stable plaques, especially in the case of atypical plaques [42]. Therefore, the application of hemodynamics is useful, and even essential, in determining carotid vulnerable plaques.

Hemodynamics characterizes the interaction between blood and the vascular wall. Sufficient evidence has shown that hemodynamic factors are involved in the formation and progression of carotid atherosclerotic plaques by causing intimal injury and inducing physiological and morphological changes in vascular endothelial cells and smooth muscle cells [43]. Commonly used hemodynamic assessment methods include computational fluid dynamics (CFD), ultrasonography (US) and 2D/3D phase-contrast MRI, but all with specific limitations [44,45]. An ideal CFD model should be based on accurate vessel geometry, blood flow boundary conditions, true blood components and vessel wall characteristics. However, in practical clinical applications, owing to the lack of patient-specific data, in order to avoid high computing costs, simplified CFD models are usually used. The accuracy of the results is limited by the accuracy of image segmentation, the selection of fluid models, the application of wall functions, the accuracy of boundary conditions and the validity of basic assumptions [46]. US is often the preferred method for measuring hemodynamics, but it often underestimates the severity because of the limitation of technical imaging. In addition, the postprocessing procedure is complex and immature [44,47]. Two-dimensional phase-contrast MRI is limited to the measurement of blood flow data on a single axial plane, and the analysis of blood flow in a certain range has limitations. Three-dimensional phase-contrast MRI lacks time-resolved compared with 4D flow. Four-dimensional flow is a time-resolved three-dimensional phase-contrast MRI technique, which can dynamically visualize carotid plaques hemodynamic changes in the entire cardiac cycle and provide a variety of hemodynamic parameters, including volume, velocity, wall shear stress (WSS), energy loss (EL) and pressure gradient (PG) [48], as shown in Table 2.

### 3.1. Hemodynamic Parameters and Formation of Carotid Plaques

First, Zhang et al. [48] measured the velocity, WSS, EL and PG at various locations of the carotid among 62 healthy volunteers aged 20–75 years and found that all the hemodynamic parameters decreased with age. Second, above all hemodynamic parameters were different at different carotid locations, which was even more pronounced at the carotid bifurcation than at other locations (Figure 3A,B). The effect of age on hemodynamics can be explained by changes in vascular elasticity and endothelial cell function with age. One reason is that vascular elasticity decreases and the vascular diameter increases with age, resulting in a decrease in blood velocity [49]. Given the association of aging and the carotid bifurcation with atherosclerotic plaque development, we think this may imply that low hemodynamic parameters are closely associated with the formation of carotid atherosclerotic plaques. This condition of hemodynamics, which resulted in plaque formation, may have been caused by a decrease in blood velocity, which lengthened the time that the blood spent in the blood vessel, causing persistent lipid retention and interaction with the vascular wall.

#### 3.1.1. WSS and Formation of Carotid Plaques

WSS has been confirmed by numerous studies as a critical hemodynamic parameter in the formation and progression of carotid atherosclerotic plaques. WSS refers to the frictional force between blood flow and the endothelium exerted on the unit area of the vascular wall. It is closely related to blood characteristics, blood velocity and vascular morphology. It is now generally accepted that low WSS is an important factor in promoting carotid plaque formation [51,52]. Arterial intima-media thickening and carotid atherosclerotic plaques initially occur at low-WSS locations (sites of vascular bifurcation and curvature) [53]. In vitro research [54] has revealed that the normal arterial WSS ranges from 1 to 7 Pa, and a WSS <1 Pa can promote the expression of atherosclerosis-related genes such as platelet-derived growth factor-A (*PDGF-A*) and vascular cell adhesion molecule-1 (*VCAM-1*); the transfer of atherosclerotic substances; and the adhesion of monocytes to the endothelium. The low WSS alters endothelial flow patterns at the molecular and cellular levels; a decrease in WSS led to the 18-fold proliferation of endothelial cells within 48 h [55]. Moreover, low WSS promotes carotid plaque formation by activating inflammatory responses [51,56], and low WSS enhanced the uptake of oxidized low-density lipoprotein, which led to an increase in the formation of lipid components in carotid plaques. All these reactions promote the development of carotid atherosclerosis [57].

#### 3.1.2. PG, EL and Carotid Plaque Formation

However, few studies have explored the relationship between PG values and carotid plaque formation. In one investigation [55], the PG values of individuals with atherosclerotic stenosis in the carotid sinus were compared to healthy individuals’ PG values, and the results revealed that the stenotic patients had lower PG values. However, this study was unable to prove that low PG values caused the development of carotid plaque, it could demonstrate only that the PG values were lower after plaque formation. Therefore, as previously mentioned, the study of Zhang et al. [48] may be more convincing because it observed that the PG values decrease with age in healthy volunteers. In addition, advanced age is one of the risk factors for carotid atherosclerotic plaque formation, which indirectly indicates that low PG values promote plaque formation. However, the association between low PG and plaque formation should also be explained by considering turbulent flow patterns and WSS rather than simply relying on age.

Similarly, there are few studies on the relationship between carotid plaque formation and EL values. Sia et al. [58] discovered that internal carotid artery stenosis may be estimated using the minimization of the EL, but whether this hemodynamic parameter influences plaque formation needs to be further confirmed.

### 3.2. Hemodynamic Parameters and Vulnerable Carotid Plaques

#### 3.2.1. WSS and Vulnerable Carotid Plaques

In addition to carotid plaque formation, WSS may also affect plaque composition and its stability. A growing amount of evidence has demonstrated that low WSS promotes carotid plaque formation, but after which, high WSS advances it into a vulnerable plaque. WSS >40 Pa can directly damage the vascular endothelium, aggravate inflammatory response and spur lipid core progression and IPH in carotid atherosclerotic plaques. The location of increased WSS frequently matches the sites of carotid plaque ulceration, inflammation and rupture [59]. Groen et al. [60] and Wu et al. [61] reported higher WSS at the ulcer site compared to the nonulcer site, whereas the plaque ulcerations in particular occur at the thicker part of the plaque. A possible explanation could be that the thicker part of the plaque more often contains IPH or LRNC, which are known to be associated with plaque rupture.

Zhang et al. [50] studied the differences in WSS between stable and vulnerable carotid plaques (classified by diffusion-weighted imaging, VW-HRMRI and clinical symptoms) in patients with moderate carotid stenosis. The reason for selecting patients with moderate stenosis (30%–70%) is that patients who undergo moderate stenosis treatments cannot be selected on the basis of the stenosis alone and require further identifying vulnerable carotid plaques [62,63]. This study found that vulnerable carotid plaques showed significantly higher WSS than stable plaques. The differences in WSS between stable and vulnerable plaques varied in different cardiac cycles and directions, which were more pronounced in the diastolic period than in the systolic period and more in the axial direction than in the circumferential direction (Figure 3C,D). The “axial” refers to the direction parallel to the centerline of the vessel and the main flow direction, and the “circumferential” refers to the direction along the circumference of lumen, orthogonal to the axial direction and centerline [64,65]. The explanation is that the diastolic period is longer than the systolic period, so the diastolic period affects the vascular wall for a longer time. Therefore, WSS has a more prominent role in diastole. Moreover, in an investigation of the connection between coronary artery axial WSS and plaque, Choi et al. [66] discovered that axial plaque stress increased with the severity of the lesion and that it had a negative correlation with the length of the lesion. These outcomes complemented one another.

#### 3.2.2. WSS and IPH, LRNC and Plaque Volume

Huang et al. found that the mean WSS at the sites of IPH was higher than that at nonbleeding sites. The reason may be that high WSS spurs the expression of vascular endothelial growth factor, which induces the formation of new vessels and damages the vascular endothelial barrier in plaques, thereby leading to IPH [67]. Tuenter et al. [68] also showed that the maximum WSS on the surface of carotid plaques was independently associated with IPH.

In addition, longitudinal and cross-sectional studies have demonstrated a link between high WSS and lipid deposition in plaques. Plaque regions of high baseline WSS become “softer” over time, i.e., lipid deposition; high WSS is related to LRNC [69].

In larger plaques (plaque burden ≥46%), a significant positive linear relationship between WSS and plaque burden was observed, but it does not indicate a causal relationship between high WSS and a higher plaque burden [70].

#### 3.2.3. WSS and Vascular Remodeling

There is controversy over the relationship between high WSS and positive vascular remodeling. In a 6-month follow-up study, Stone et al. observed that regions of low baseline WSS exhibited positive vascular remodeling [71]. However, an 8-month follow-up study related low baseline WSS to negative vascular remodeling [72]. Samady et al. [73] showed a consistent observation: regions of high baseline WSS were more likely to develop positive vascular remodeling. Park et al. [74] found that regions of high WSS had a higher percentage of positively remodeled plaques compared with regions of low and moderate WSS. Positive vascular remodeling is an imaging characteristic of vulnerable plaques, thus suggesting that high WSS is associated with vulnerable plaques.

#### 3.2.4. PSS, PG and Vulnerable Carotid Plaques

Plaque structural stress (PSS) is deemed to play a potential role in carotid plaque rupture. When PSS exceeds the strength of the fibrous cap covering the plaque, plaque rupture occurs. Histology research revealed higher PSS in the area of plaque rupture [75]. In addition, in vivo carotid artery research showed higher PSS in vulnerable plaques than in stable plaques [76]. Higher WSS plaque regions were associated with higher PSS, suggesting that high WSS and high PSS work together to promote plaque progression up to a certain threshold, where the plaque ruptures, leading to AIS [77]. Moreover, one study found that PG might play a more important role than WSS in plaque rupture; however, more research is needed to confirm this [45].

## 4. Radiomic Research on Vulnerable Plaques

Although VW-HRMRI and 4D flow can provide accurate qualitative and quantitative assessments of the morphology and hemodynamic of carotid atherosclerotic plaques, both the qualitative and quantitative characteristics are subject to the subjectivity of observers and the heterogeneity of imaging protocols. At the same time, substantial microcosmic information is missing. Machine learning based on radiomic analysis provides a new method to quantify the heterogeneity of lesions through high-throughput extraction and an analysis of digital features from medical images by using computer software. It has been used in the qualitative diagnosis of lesions, clinical grading and staging, tumor genetic biomarker identification, efficacy evaluation and prognosis. The radiomics research includes the following steps: (1) image acquisition and preprocessing; (2) regions of interest (ROIs) segmentation; (3) feature extraction and screening; and (4) model establishment and model performance evaluation. Initial radiomic studies on atherosclerotic plaques were based on a texture analysis of CT or US images [78,79]. Compared with CT and US, VW-HRMRI is advantageous in high soft-tissue contrast and multisequence imaging, which may provide more valuable information. However, only three radiomic studies based on VW-HRMRI atherosclerotic plaques have been reported to date.

### 4.1. Radiomic Research on Basilar Artery Plaques

Shi et al. [80] conducted a radiomic study that was based on VW-HRMRI to distinguish stable and vulnerable basilar artery plaques. The authors used 3D-Slicer software to manually delineate ROIs at a slice of the maximum plaque area on T1WI, T2WI and T1WI-CE images. In total, 94 radiomic features were extracted, and the features with significant differences (*p* < 0.05) and the AUCs >0.65 were set as the inputs for random forest training features. The results showed that seven radiomic features from T1WIs and three from T1WI-CE images were independently associated with plaque stability. The diagnostic performance parameter AUC for plaque classification was 0.833 with VW-HRMRI features, 0.893 with T1WI radiomic features and 0.918 with T1WI-CE radiomic features. The combination of T1WI and T1WI-CE radiomic features improved the AUC to 0.936, significantly higher than those of the imaging feature model and single-sequence radiomic feature model.

### 4.2. Radiomic Research on Middle Cerebral Artery and Basilar Artery Plaques

Subsequently, Shi et al. [81] performed a histogram texture analysis that was based on VW-HRMRI to extract the first-order texture features of atherosclerotic plaques in the middle cerebral artery and basilar artery, and they explored the differences in histogram features between stable and vulnerable plaques. The results showed that the histogram-defined coefficient of variation (CV) on T1WIs was an independent predictive parameter for classifying the type of plaques, where the sensitivity, specificity and accuracy were 0.79, 0.80 and 0.80, respectively. The CV of vulnerable plaques was greater than that of stable plaques. However, the ability of CV to distinguish vulnerable and stable plaques was demonstrated only on T1WI, not on T2WI and T1WI-CE. Although the exact mechanism is unclear, the reason may lie in the composition of atherosclerotic plaques. IPH, LRNC, calcification and fibrous tissue components in plaques lead to varying signal intensities on T1WIs.

### 4.3. Radiomic Research on Carotid Artery Plaques

Zhang et al. [12] established a high-risk carotid plaque model based on MRI radiomic features and evaluated its performance in distinguishing stable and vulnerable carotid plaques relative to the model based on traditional MRI features. The traditional model was built using IPH and LRNC features derived from multivariable logistic regression analysis. The authors used the open-source software ITK-SNAP to manually draw ROIs at a slice of the maximum plaque area on T1WI, T2WI, DCE and delayed DCE images for radiomic analysis. Those MRI sequences obtained 85, 68, 72 and 162 radiomic features, respectively. In total, 33 radiomic features were finally retained through the Lasso algorithm for constructing a radiomic model. The results showed that for the training and testing sets, the AUCs of the traditional model were 0.825 and 0.804, respectively; the AUCs of the radiomic model were 0.988 and 0.984, respectively; and the AUCs of the combined model were 0.989 and 0.986, respectively. It can be seen that the radiomic model and combined model showed better performance than the traditional model, but no significant difference was observed between the radiomic model and combined model.

These results indicate that compared with the traditional subjective qualitative and quantitative imaging characteristics, the radiomic method enables finding more differential features, showing its higher value in determining plaque vulnerability. However, the ROIs in the above studies are all based on manual segmentation, which is time-consuming and subject to observer variability. Furthermore, complex vascular curvature, too-small plaques and surrounding tissue disturbance pose more challenges to the manual segmentation of plaques.

## 5. Automatic Segmentation of Carotid Atherosclerotic Plaques

At present, artificial intelligence has played an indispensable role in the medical industry. Plaque automatic segmentation based on artificial intelligence can improve the work efficiency of physicians. So far, various segmentation methods have been proposed at home and abroad, which are predominantly divided into those based on traditional image processing and those based on deep learning. The segmentation results based on traditional image processing are not accurate enough to achieve the requirements of clinical diagnosis.

Deep learning was first proposed in 2006 by Hinton, who then gradually applied it in image processing [82]. Deep-learning segmentation methods are mainly convolutional neural networks (CNN) and other CNN-derived networks. Earlier CNN-based image segmentation methods used a sliding window to extract small image patches and input them into CNN networks. The sliding window is much smaller than the whole image. Therefore, so many image patches need prediction for an image that the computational efficiency is reduced. At the same time, the sliding window is so small that only limited local features can be extracted. Given these problems, Long et al. [83] in 2015 proposed fully convolutional networks (FCN), which can use any size of image for segmentation. However, owing to the great differences between medical images and natural images, the segmentation network suitable for natural images shows poor performance for medical images. Thus, Ronneberger et al. [84] in 2015 proposed a segmentation network based on FCN for medical images, called U-Net, which has become the most common segmentation architecture in the medical field. The original U-Net can segment plaque morphology, but it has over-segmentation and under-segmentation problems. U-Net combines the optimization strategy and has been optimized at the network, loss function, heuristic rules, preprocessing and other levels, and the segmentation effect is significantly improved. At present, many researchers have devoted themselves to the study of carotid atherosclerotic plaque segmentation, but their studies are mainly based on US images, as shown in Table 3.

Few studies have been reported on VW-HRMRI plaque segmentation. In 2017, Dong et al. [88] used FCN as the basic network architecture and GoogLeNet, VGG-16 and ResNet-101 as the feature extraction networks to segmentate carotid plaque components (calcification, lipid core, hemorrhage, loose matrix, etc.) on 2D images of MRI four sequences, with segmentation accuracy reaching up to 0.56. Based on MRI images, Peng D et al. [89] optimized U-Net with pyramid pooling and global feature attention upsampling and squeeze-and-excitation mechanisms, which produced superior segmentation to ordinary U-Net with an 8% increase in the Dice coefficient. Moreover, for small-sized plaques and small sample datasets, a study adopted an image patch–based acquisition strategy for data amplification, introduced a transformer model to the segmentation network and proposed a 3D Trans-IS U-Net plaque segmentation algorithm. The results showed that the segmentation algorithm accuracy for carotid plaques reached 0.74, significantly higher than other segmentation networks [90].

Although the above automatic segmentation accuracies are not satisfactory, these studies have proven that it is feasible to apply neural networks to automatically segment carotid plaques and their components. It lays the foundation for future research and provides new research directions.

## 6. Conclusions

In summary, VW-HRMRI, 4D flow and artificial intelligence improve the efficiency and accuracy of vulnerable plaque assessment, which are significant for guiding clinical aggressive therapy to reduce the risk of plaque rupture and stroke. However, VW-HRMRI has high requirements for hardware and software, and the inspection time is long. At present, the research and application of 4D flow are still few, and the major hardware manufacturers have not yet fully commercialized VW-HRMRI. Radiomics studies on atherosclerotic plaques are retrospective at present, which cannot determine the ability to predict the risk of stroke. Therefore, large-sample multicenter prospective studies are needed to provide more valuable information in future and ultimately promote these advanced technologies so that they may be used in efficient clinical applications.

## Figures and Tables

**Figure 1 brainsci-13-00143-f001:**
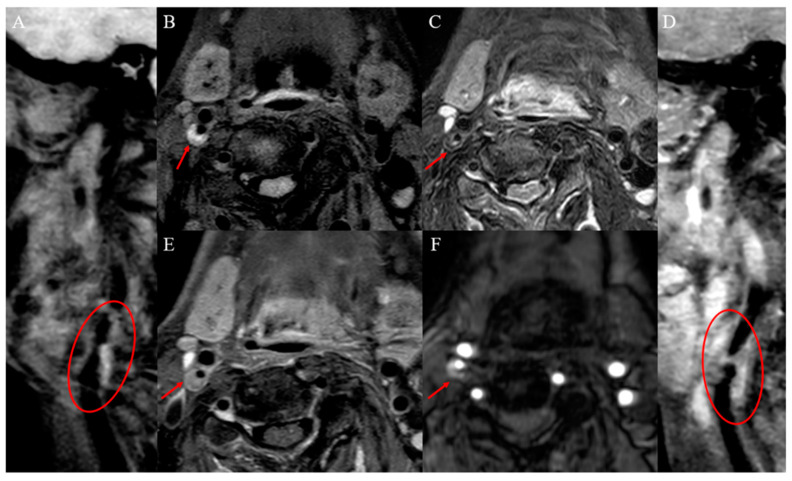
Definition of vulnerable carotid plaque (IPH and incomplete fibrous cap). Pre-enhanced VW-HRMRI 3D T1WI (**A**,**B**): inhomogeneous hyperintensity in the plaque and plaque surface was not smooth. T2WI-SPAIR image (**C**): hypointensity in the plaque. Post-enhanced VW-HRMRI 3D T1WI (**D**,**E**): discontinuous thin-linear enhancement on plaque surface. TOF image (**F**): hyperintensity in the plaque.

**Figure 2 brainsci-13-00143-f002:**
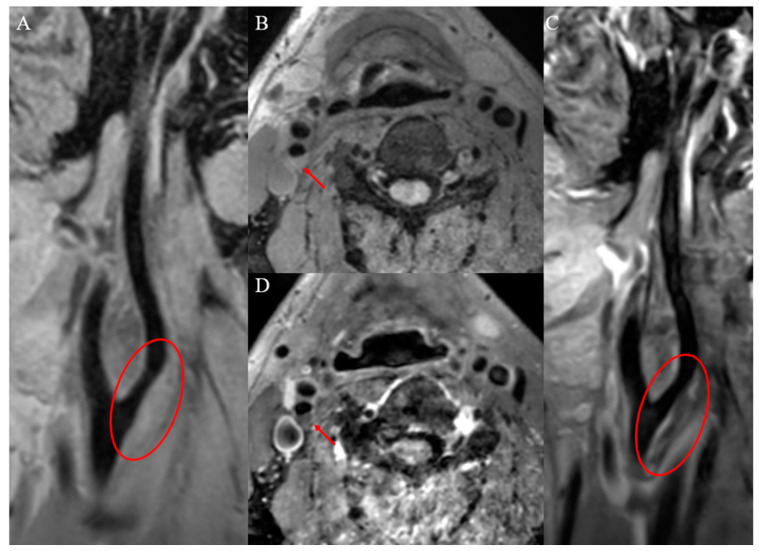
Definition of stable carotid plaque (intraplaque lipid and complete fibrous cap). Pre-enhanced VW-HRMRI 3D T1WI (**A**,**B**): homogeneous isointensity in the plaque and plaque surface was smooth. Post-enhanced VW-HRMRI 3D T1WI (**C**,**D**): continuous thick-linear enhancement on plaque surface.

**Figure 3 brainsci-13-00143-f003:**
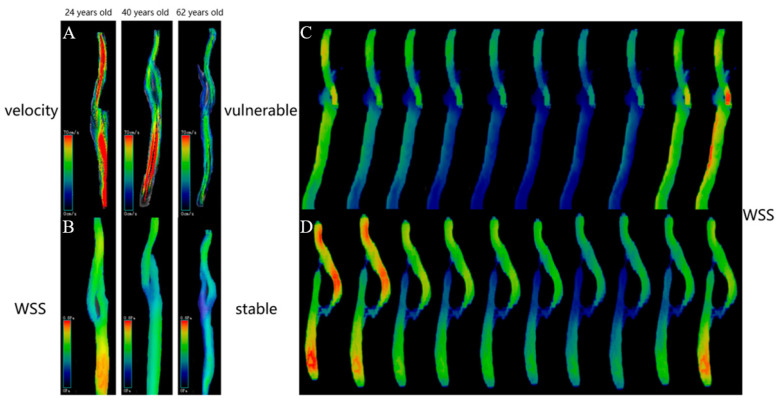
Examples of hemodynamic parameters (velocity and WSS) are visibly shown by 4D flow. (**A**,**B**): these cases showed that the velocity and WSS decreased with age in healthy volunteers’ carotids, and the proximal internal carotid artery decreased significantly [48]. (**C**,**D**): the WSS in the vulnerable carotid plaque increased and the stable carotid plaque declined, and the degree of changes varied at different time points [50].

**Table 1 brainsci-13-00143-t001:** The definition and calculation formula of the quantitative indicators of carotid plaques.

Quantitative Indicators	Definition and Calculation Formula
Plaque volume	Total plaque volume = total wall area × (slice thickness + slice gap)
Plaque thickness	Plaque thickness = (the diameter of the outer wall of the lumen diameter of the inner wall of the lumen) at the plaque level.
Plaque normalized wall index (NWI)	NWI = wall area (WA)/[lumen area (LA) + wall area (WA)]
Plaque remodeling index (RI)	RI = vascular area at the slice with the narrowest lumen/vascular area at the adjacent slice with normal vascular wall
Vascular stenosis ratio	Stenosis ratio = (1 – [lumen area at the slice with the narrowest lumen/lumen area at the adjacent slice with normal vascular wall]) × 100%

**Table 2 brainsci-13-00143-t002:** Hemodynamic parameters and significance.

Parameters	Significance
Volume	Blood flow volume through the section in unit time
Velocity	Blood flow velocity through the section in unit time
Wall shear stress (WSS)	The force of blood flow on the vessel wall along the tangent direction of the vessel
Energy loss (EL)	Ratio of lost energy to vessel volume in deformation cycle
Pressure gradient (PG)	Pressure change per unit distance

**Table 3 brainsci-13-00143-t003:** Application based on U-Net deep-learning algorithm in carotid atherosclerotic plaque segmentation with US images.

References	Segmentation Algorithm	3D	Number of Samples	Segmentation Accuracy
Meshram et al. [85] (2020)	U-Net	No	101	Automatic: 0.48 Semiautomatic: 0.83
Meshram et al. [85] (2020)	Dilated U-Net	No	101	Automatic: 0.55 Semiautomatic: 0.84
Xie et al. [86] (2020)	U-Net	No	226	0.67
Xie et al. [86] (2020)	Dual-decoder convolutional U-Net	No	226	0.69
Jiang et al. [87] (2020)	Three-direction U-Net	Yes	22	0.68

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
