# Peer review of "Imaging and Hemodynamic Characteristics of Vulnerable Carotid Plaques and Artificial Intelligence Applications in Plaque Classification and Segmentation"

_brainsci, 2023, doi:10.3390/brainsci13010143_

Round 1
Reviewer 1 Report
This review paper summarized developments in imaging and hemodynamic characteristics of vulnerable carotid plaques based on VW-HRMRI and four-dimensional (4D) flow MRI, and discussed the relationship between these characteristics and ischemic stroke. In addition, the applications of artificial intelligence in plaque classification and segmentation were reviewed. This article is well written overall. However, for publication, the following concerns should be addressed.
1. Animal studies and clinical trials such as NASCET have been shown that the dents of plaque surface ulceration create a vortex flow that promotes platelet aggregation, resulting in an increased risk of future stroke through a mechanism such as artery to artery embolism. In this regard, a more in-depth investigation of plaque surface ulceration is required.
2. How about providing an additional figure to intuitively show the definition and calculation formula of the quantitative indicators included in “1.2. Quantitative analysis of vulnerable carotid plaques based on VW-HRMRI”?
3. As the author described, VW-HRMRI and 4D flow seem superior to other modalities in evaluating vulnerable carotid plaques. Nevertheless, there must be limitations of VW-HRMRI and 4D flow. These points should also be included in the text.
4. There are some typing errors.
5. What are the specific limitations of CFD, US, and 2D/3D phase-contrast MRI in hemodynamic analysis?
6. What formulas are used to derive the energy loss (EL), pressure gradient (PG), and plaque structural stress (PSS)? What do these low or high values ​​mean hemodynamically?
7. The association between low PG and plaque development should be explained by considering turbulent flow pattern and WSS rather than simply relying on age. What do the authors think?
Reviewer 2 Report
I would like to congratulate the authors on their work! This is potentially significant review regarding the developments in imaging and hemodynamic characteristics of vulnerable carotid plaques based on VW-HRMRI and 4D flow. Moreover, the authors discussed the relationship between these characteristics and ischemic stroke and the applications of artificial intelligence in plaque classification and segmentation.
Well done.
Author Response
Thank you for your spended precious time reviewed the manuscript, thank you for your recognition and affirmation of our work, and best wishes to you!
Reviewer 3 Report
Vulnerable plaque and its imaging and assessment are hot research topic. The rupture of vulnerable carotid atherosclerotic plaques is the most common cause of acute ischemic stroke (AIS) across the world. Currently, vessel wall high-resolution magnetic resonance imaging (VW-HRMRI) is the most appropriate and cost-effective imaging technique to characterize carotid plaque vulnerability and plays an important role in promoting early diagnosis and guiding clinical aggressive therapy to reduce the risk of plaque rupture and AIS. In recent years, great progress has been made in imaging research on vulnerable carotid plaques. This review summarized developments in imaging and hemodynamic characteristics of vulnerable carotid plaques based on VW-HRMRI and four-dimensional (4D) flow MRI, and discusses the relationship between these characteristics and ischemic stroke. In addition, the applications of artificial intelligence in plaque classification and segmentation are reviewed.
This is a very interesting paper reviewing an area with important public health significance. Carotid plaque imaging and flow calculations were reviewed. However, plaque imaging and flow imaging are not always tied together. Due to clinical needs and insurance coverages, MRI are often down without flow quantification. MRI techniques to detect and quantify plaque compositions were developed and great progress have been made there. Those work should be covered in this review (Chun Yuan et al.). Carotid modeling with flow only, fluid-structure interactions, and with linkage to plaque progression and vulnerability assessment and predictions should also be reviewed.
